# Concentrations and fluxes of suspended particulate matters and associated contaminants in the Rhône River from Lake Geneva to the Mediterranean Sea

Hugo Lepage[1], Alexandra Gruat[2], Fabien Thollet[2], Jérôme Le Coz[2], Marina Coquery[2], Matthieu Masson[2], Aymeric Dabrin[2], Olivier Radakovitch[1], Jérôme Labille[3], Jean-Paul Ambrosi[3], Doriane Delanghe[3], Patrick Raimbault[4]

[1] Institut de Radioprotection et de Sureté Nucléaire (IRSN), PSE-ENV, SRTE/LRTA, BP 3, 13115 Saint-Paul-lez-Durance, France
[2] INRAE, UR RiverLy, 5 Rue de la Doua CS 20244, Villeurbanne F-69625, France
[3] Aix-Marseille Univ, CNRS, IRD, INRA, Coll France, CEREGE, 13545 Aix-en-Provence, France
[4] Aix Marseille Université, CNRS/INSU, Université de Toulon, IRD, Mediterranean Institute of Oceanography (MIO), UM110, 13288 Marseille, France

*Correspondence to*: Hugo Lepage (hugo.lepage@irsn.fr)

**Abstract.**

The Rhône River is amongst the main rivers of Western Europe and the biggest by freshwater discharge and sediment delivery to the Mediterranean Sea. Its catchment is characterized by distinct hydrological regimes that may produce annual sediment deliveries ranging from 1.4 to 18.0 Mt y$^{-1}$. Its course meets numerous dams, hydro- and nuclear power plants, and agricultural, urban or industrial areas. Moreover, with the climatic crisis we are currently facing, it is proven that the occurrence and the intensity of extreme events (floods or droughts) will increase. Therefore, it is crucial to monitor the concentrations and fluxes of suspended particulate matters (SPM) and associated contaminants to study the current trends and their evolution. In the Rhône River (from Lake Geneva to the Mediterranean Sea), a monitoring network of 15 stations (three on the Rhône River and 12 on tributaries) has been set up in the past decade by the Rhône Sediment Observatory (OSR) to investigate the concentrations and the fluxes of SPM and associated contaminants, as well as their sources. A main purpose of this observatory is to assess the long term trend of the main contaminant concentrations and fluxes, and to understand their behavior during extreme events such as floods or dam flushing operations. The dataset presented in this paper contains the concentrations and fluxes of SPM as well as the concentrations and fluxes of several particle bound contaminants of concern (PCB, TME, radionuclides), the particle size distribution and the particulate organic carbon of SPM. Sediment traps or continuous flow centrifuges were used to collect sufficient amount of SPM in order to conduct the measurements, and data completion was applied to reconstruct missing values. This observatory is on-going since 2011 and the database is regularly updated. All the data are made publicly available in French and English through the BDOH/OSR database at https://doi.org/10.15454/RJCQZ7 (Lepage et al., 2021).

## 1 Introduction

Human activities, especially in recent decades, impact rivers all over the world and consequently the seas and oceans. This impact is enhanced by the climate changes we are experiencing. In order to understand these changes in our rivers, and to better anticipate them, it is important to set up the monitoring of the water and particles transported by these environmental vectors (Syvitski et al., 2005). In addition, the monitoring of rivers allows a better understanding of changes in the seas and oceans whose monitoring is more difficult to implement (Vihma et al., 2019). While monitoring of the quantity of water in transit has been effective for several decades (Horowitz, 2008), there are fewer examples of long-term monitoring of SPM and contaminants (Syvitski et al., 2005), leading to many gaps in their behavior and fate on large time scales. Indeed, the literature contains studies on large rivers (Horowitz et al., 2001; Armijos et al., 2017; Martinez et al., 2013), but many other studies are based on either low frequency sampling/measurement over a long period (Lick, 2008; Delmas et al., 2012; Moatar et al., 2013), or high frequency sampling/measurement over a short period (Radakovitch et al., 2008; Sicre et al., 2008; Panagiotopoulos et al., 2012). It is crucial to extend the observations in different watersheds over the world to better constrain the hydro-sedimentary dynamics according to variables which have a direct relationship with the production of sediments and associated contaminants such as vegetation cover, industrialization, population density or agriculture. Moreover, contaminants are mainly measured in the liquid fraction of samples, which is analytically simpler than measuring them in the solid fractions. However, the transport of contaminants by the solid fraction is far from negligible (Horowitz, 2009). To better understand the global changes that affect our terrestrial and marine environments, it is therefore important to develop long-term hydro-sedimentary observatories.

The Rhône River (813 km long) rises in the Rhône Glacier in the Swiss Alps, transits through Lake Geneva and flows through southeastern France down to the Mediterranean Sea. It is one of the biggest Mediterranean rivers in terms of freshwater and suspended particulate matter (SPM) delivery to the sea (Ludwig et al., 2009; Sadaoui et al., 2016). Several exceptional floods occurred in the last two decades and have modified its morphology (Antonelli et al., 2008). In fact, the annual SPM flux near the outlet strongly varies due to distinct hydrological regimes in the basin, including glacial, nival, pluvial and Mediterranean components (Pont et al., 2002). Between 1.4 Mt to 18.0 Mt of sediment transit to the Mediterranean Sea each year (Poulier et al., 2019), while the mean inter-annual monthly SPM flux is characterized by a tri-modal distribution over the year with maxima centered in November, January, and May-June (Delile et al., 2020). The Rhône channel is widely artificial with 21 hydroelectric dams, five nuclear power plants and two big cities with over 500 000 inhabitants (Geneva in Switzerland and Lyon in France). The Rhône River is an important water resource at the inter-regional scale, notably for drinking water supplies or irrigation as a large area of this catchment is used for agriculture, especially farming and grazing. The Rhône watershed covers about a $5^{th}$ of the surface of metropolitan France, which implies the transport of eroded material from a wide variety of land uses. Therefore, anthropic contamination of the Rhône River by hydrophobic organic contaminants such as polychlorobiphenyls (PCBs) and polycyclic aromatic hydrocarbons (PAHs), or by trace metal elements (TME), or radionuclides has been observed for many years (Radakovitch et al., 2008; Mourier et al.,

2014; Delile et al., 2020; Eyrolle et al., 2020). For these substances, SPM transport represents the main driver of contaminants from rivers to coastal areas, leading to an alteration of bio-geochemical cycles and water quality (Horowitz, 2009).

In this watershed, studies conducted on sediment dynamics and associated contaminants are unfortunately scarce (Antonelli et al., 2008; Radakovitch et al., 2008; Panagiotopoulos et al., 2012; Delmas et al., 2012) and do not allow to understand the

observed changes over the long term. On this basis, the monitoring of spatial and temporal distribution of SPM and associated contaminants has been conducted within the Rhône Sediment Observatory (OSR) since 2009 (Le Bescond et al., 2018). Fifteen monitoring stations have been installed with 3 of them along the Rhône River channel and 12 on the main tributaries. This monitoring network was designed to improve our understanding on SPM transfer processes in rivers exposed to anthropogenic contamination and extreme hydro-sedimentary events (flood, low-water, dam regulation), and to

provide stakeholders with precise values of SPM and contaminants fluxes. The current database provides time series of Suspended Solid Concentrations (SSC), SPM fluxes, Particle Size Distributions (PSD), Particulate Organic Carbon (POC) contents, and the concentrations and fluxes of several contaminants of interest (Thollet et al., 2021).

The hydro-sedimentary and contaminant monitoring data from the Rhône River and the main tributaries are useful for the assessment of:

- The annual and inter-annual fluxes of SPM and hydrophobic organic contaminants and TME and radionuclides;
- The spatial and temporal variations of the contaminant concentrations;
- The impact of extreme events (floods or dam flushing operations) in term of sediment budget and contaminant concentrations.

## 2 Description of the data and the functionality of the database (BDOH/OSR)

In the database (https://doi.org/10.15454/RJCQZ7, Lepage et al. (2021)), two types of time series are stored. Discontinuous time series are used for measurements on SPM samples that are collected from a start time to an end time (with no information in between sampling periods). Calculated time series are obtained by several transformations (including data completion) of the discontinuous time series and the SSC (Fig. 1). Time series can be interpolated, transformed and multiplied within the database application to derive computed time series from existing time series.

The dataset includes the following measured parameters (discontinuous time series):
- Particle Size Distribution - PSD (10%, 50%, 90% percentile diameters D10, D50, D90 - in µm);
- Particulate Organic Carbon - POC (in g kg$^{-1}$ dry weight);
- Trace metal elements (TME): arsenic (As), cadmium (Cd), chromium (Cr), cobalt (Co), copper (Cu), lead (Pb), mercury (Hg), nickel (Ni), zinc (Zn) (in mg kg$^{-1}$ dry weight);

- Polychlorobiphenyls (PCB) : PCB28, 52, 101, 118, 138, 153, 180 (in µg kg$^{-1}$ dry weight);
- Radionuclides: Organically Bound Tritium (OBT - in Bq kg$^{-1}$ dry weight), Carbon-14 ($^{14}$C - in Bq kg$^{-1}$ dry weight of carbon) and Cesium-137 ($^{137}$Cs - in Bq kg$^{-1}$ dry weight).

The dataset includes the following calculated parameters:

- Suspended Solid Concentration - SSC (mg L$^{-1}$) either derived from in-situ turbidity measurements or by filtration of water samples as explained in the Method section. The procedure for data completion are presented in the Method section,
- SPM flux (t s$^{-1}$) was calculated in the database by multiplying water discharge (not made available in the dataset because it is released separately by data producers - in m$^3$ s$^{-1}$) and SSC (Fig. 1). The owners of the water discharge data are reported in Tables 2 and 3,
- Contaminant flux (g s$^{-1}$ or Bq s$^{-1}$) was calculated in the database by multiplying the SPM flux and the contaminant concentration (Fig. 1). The procedures for flux computation and data completion of the discontinuous contaminant time series are presented in the Method section.

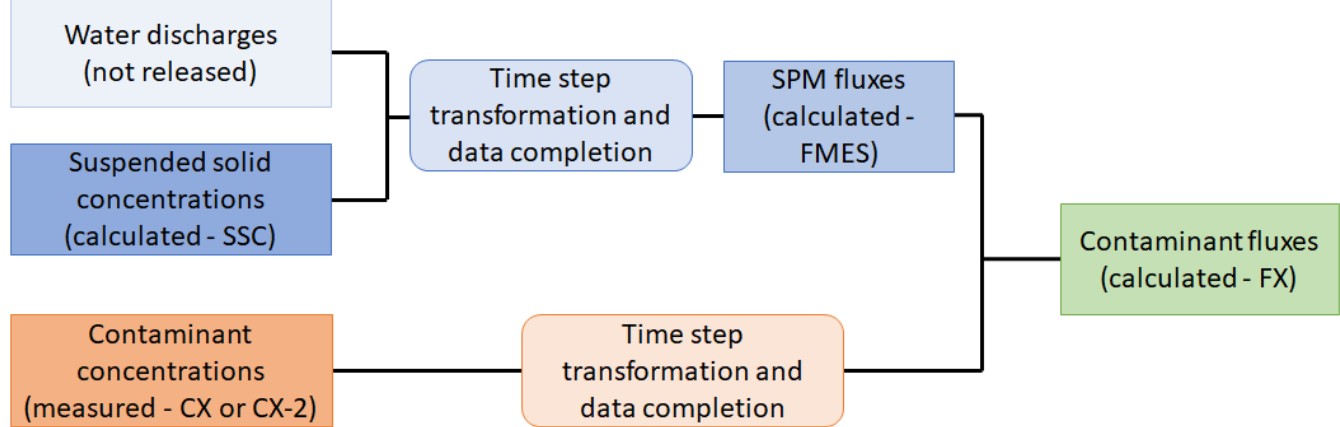

**Fig. 1 - Calculation processes to obtain SPM and particulate contaminant fluxes from the measured parameters (where X is a contaminant).**

Data are organized by station with information on the provider, the period of availability (date in Coordinated Universal Time – UTC) and the number of data (Fig. 2A). For particulate contaminant concentration, the name of the time series is related to the sampling method, as illustrated in Fig. 2A for Hg, with:

- CX = discontinuous concentration of particulate contaminant X in SPM samples collected by continuous flow centrifuge,
- CX-2 = discontinuous concentration of particulate contaminant X in SPM samples collected by particle trap.

**A**

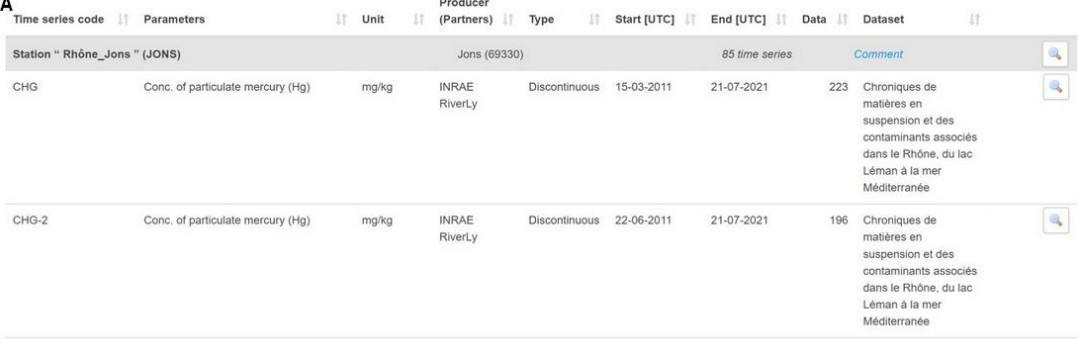

| Time series code | Parameters | Unit | Producer (Partners) | Type | Start [UTC] | End [UTC] | Data | Dataset | |
|---|---|---|---|---|---|---|---|---|---|
| Station " Rhône_Jons " (JONS) | | | Jons (69330) | | | 85 time series | | Comment | 🔍 |
| CHG | Conc. of particulate mercury (Hg) | mg/kg | INRAE RiverLy | Discontinuous | 15-03-2011 | 21-07-2021 | 223 | Chroniques de matières en suspension et des contaminants associés dans le Rhône, du lac Léman à la mer Méditerranée | 🔍 |
| CHG-2 | Conc. of particulate mercury (Hg) | mg/kg | INRAE RiverLy | Discontinuous | 22-06-2011 | 21-07-2021 | 196 | Chroniques de matières en suspension et des contaminants associés dans le Rhône, du lac Léman à la mer Méditerranée | 🔍 |

**B**

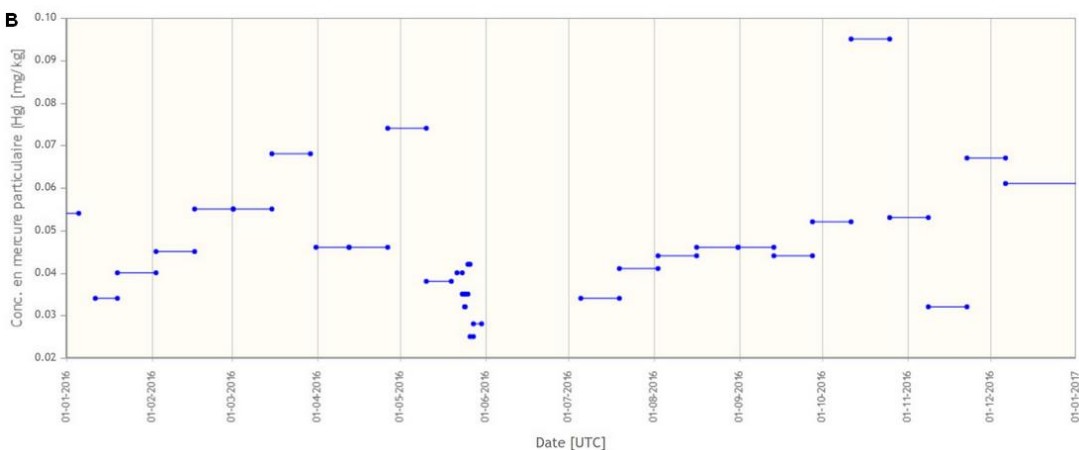

**C**

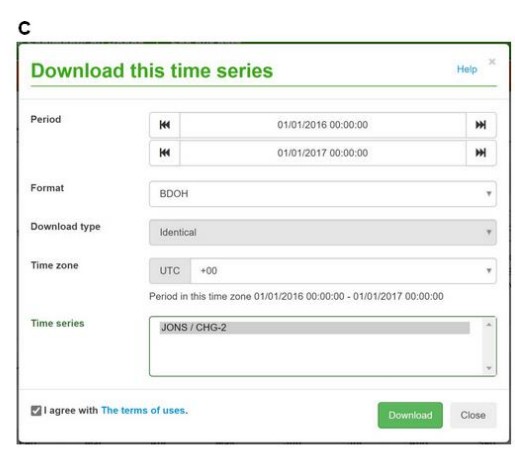

**D**

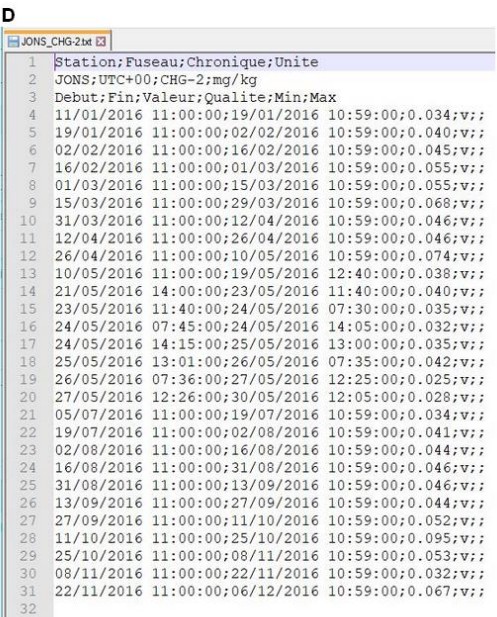

**Fig. 2 – Example of time series available at the Rhône station at Jons in the BDOH/OSR database: A) Screenshot of mercury (Hg) time series with CHG for SPM sample collected by continuous flow centrifuge and CHG-2 for SPM sample collected by particle trap, B) Screenshot of the visualization of the time series CHG-2 in 2016 , C) Screenshot of the window for choosing download parameters of the time series CHG-2 and D) Screenshot of the flat text file of the time series CHG-2 in 2016 and associated quality code (v = valid).**

The user can visualize each time series online (Fig. 2B) and download them individually after choosing several parameters (Fig. 2C) such as:

- the period of the time series by selecting start and end dates;
- the file format: BDOH (raw) or Hydro2-QTVAR (specific format used by the French national hydrological services);
- the type of transformation and time steps: identical (raw data, constant or variable time steps), linear interpolation at constant time steps, mean (1 or 6 hours, daily, monthly, yearly, event scale) or accumulation (1 or 6 hours, daily, monthly, yearly, event scale). This parameter is only available for calculated and continuous time series;
- the time zone (by default in the database: UTC +00).

Data are sent by e-mail in a compressed folder (export.zip) containing the time series as flat text files (STATION_PARAMETER.txt) and another file (Report.txt) that contains all the necessary metadata (producer, parameter, genealogy , time zone, conversion factors). The file of the time series can be processed by any software (Excel, R, Matlab, etc.). Each value of the time series is associated with a quality code (Fig. 2D) according to Table 1.

**Table 1 - Quality codes used in the BDOH database**

| Code | Status | Description |
|------|--------|-------------|
| v | valid | the value is accurately quantified and coherent with the other values of the parameters |
| a | missing information | the quality of the data cannot be assessed |
| l | missing value | missing value due to logistic problem. By default, the value associated is -9999 |
| i | invalid | outlier value that was removed |
| d | questionable | the value is accurately quantified but is not coherent with the other values of the parameters |
| e | estimated value | the value (coded "l" or "i") was estimated or modeled following the method described in the Method section . |
| lq | limit of quantification | the value is lower than the limit of quantification |
| ld | limit of detection (used for radionuclides only) | the value is lower than the limit of detection |

## 3 Methods

### 3.1 Sampling location

The monitoring conducted within the OSR is located in the Rhône River catchment downstream of Lake Geneva in Switzerland (~95 000 km²) (Fig. 3). Due to its large volume and high trapping efficiency, Lake Geneva is a barrier to SPM

coming from the upstream catchments so that its sedimentary output can be neglected. Fifteen monitoring stations have been installed (Fig. 3) with three of them along the Rhône River (Table 2) and 12 on the main tributaries (Table 3).

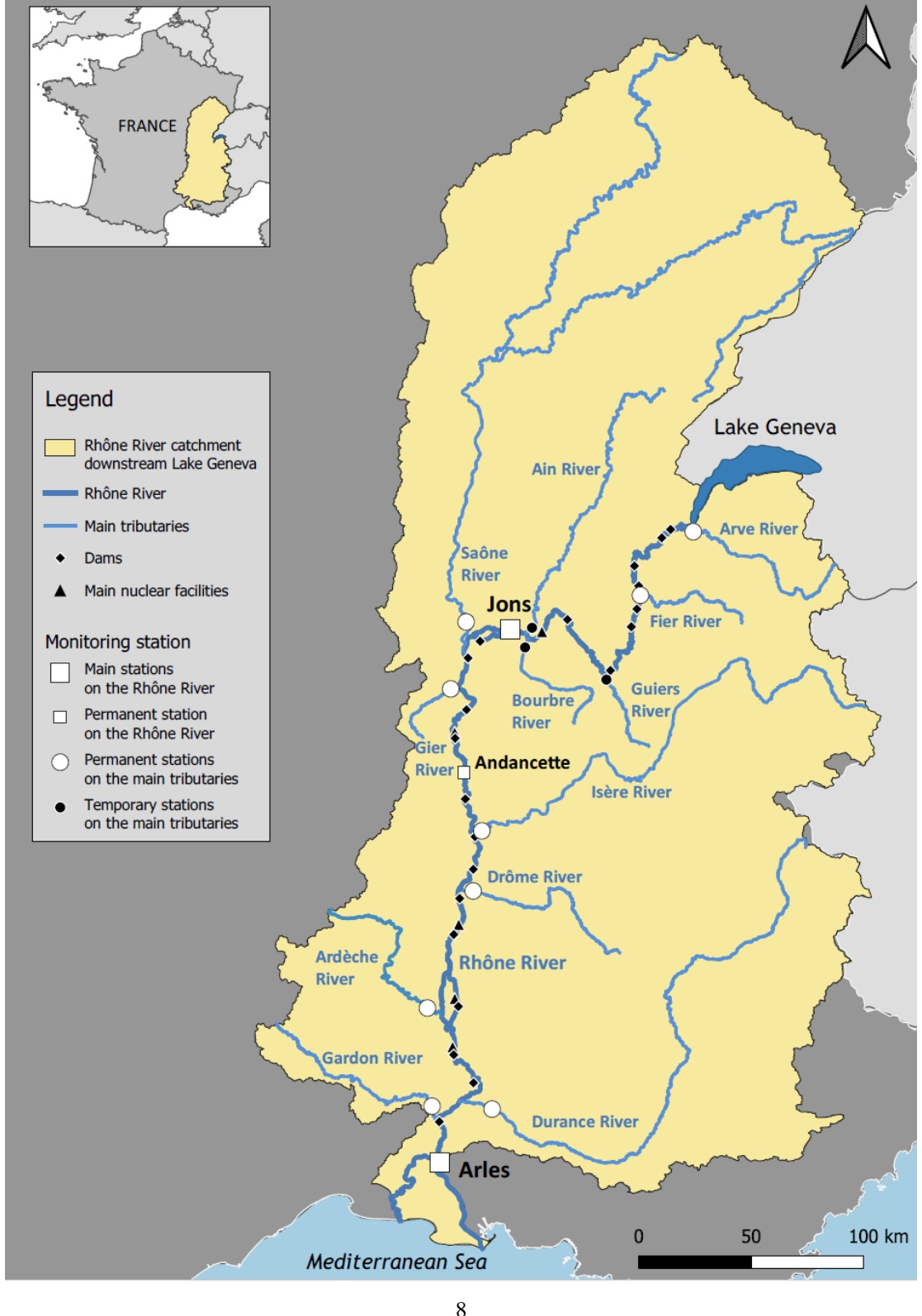

**Fig. 3 – Location of the OSR monitoring stations along the Rhône River and at the outlets of the main tributaries, from Lake Geneva to the Mediterranean Sea.**

**Table 2 – Location of the sampling stations on the Rhône River (from upstream to downstream). CNR: Compagnie Nationale du Rhône, DREAL: Direction Régionale de l'Environnement, de l'Aménagement et du Logement, INRAE: Institut National de Recherche pour l'Agriculture, l'alimentation et l'Environnement, MIO: Mediterranean Institute of Oceanology, VNF : Voies Navigables de France.**

| Location | Description | Status | Location of the Water discharge measurement (WGS84 coordinates, provider) | Location of the SSC measurement (WGS84 coordinates, provider) | Location of the SPM sampling station (WGS84 coordinates) |
|---|---|---|---|---|---|
| Jons | Reference station to evaluate concentrations and fluxes from the Upper Rhône River, and upstream of the city of Lyon. | Permanent | Computed using a 1D hydrodynamical model (Launay et al., 2019) requiring the discharge data of the Rhône River at the station V1630020 at Lagnieu (45.8814226, 5.3404402 - CNR), and those from the Ain at the station Port Galland (45.8163239, 5.2133034 - CNR) and the Bourbre at the station V1774010 (45.7152493, 5.1591093 – DREAL Auvergne-Rhône-Alpes). | Station of the Grand Lyon (45.811884, 5.086006 – Grand Lyon/Véolia/INRAE) | 45.811884, 5.086006 |
| Andancette/Saint-Vallier | Intermediate station in the Rhône River, between the city of Lyon and the confluence with the Isère river. | Permanent | Computed discharge at Gervans hydropower station (45.1095485, 4.8219966 - CNR) and Arras-sur-Rhône dam (45.136558, 4.807434 - CNR) | Station of CNR at Saint-Vallier 45.183414, 4.813635 | 45.243563, 4.802822 |
| Arles | Reference station to evaluate concentrations and fluxes near the outlet in the Mediterranean Sea. | Permanent | Station at Arles, PK 282.650) (43.7877075, 4.6528746 - CNR/VNF) | Rhône Observatory Station at Arles (43.678750, 4.621139 - MIO) | 43.678750, 4.621139 |

**Table 3 – Description of the monitored tributaries and location of the sampling stations. CNR: Compagnie Nationale du Rhône, DREAL: Direction Régionale de l'Environnement, de l'Aménagement et du Logement, EDF: Electricité de France, INRAE: Institut National de Recherche pour l'Agriculture, l'alimentation et l'Environnement, MIO: Mediterranean Institute of Oceanology, FOEN: Swiss Federal Office for the Environment, SIG: Services Industriels de Genève. NA = not available**

| Basin (area, mean annual water discharge, mean of the total annual SPM | Description | Status | Location of the Water discharge measurement (WGS84 coordinates, provider) | Location of the SSC measurement (WGS84 coordinates, provider) | Location of the SPM sampling station (WGS84 coordinates) |
|---|---|---|---|---|---|

flux )

| River | Description | Type | Station | Station |
|---|---|---|---|---|
| Arve (2 083 km², 74 m³ s⁻¹, 0.56 ± 0.20 Mt) | Steep mountain catchment with high flow in spring that provides most of the SPM in the upstream Rhône catchment. | Permanent | Station 2170 at Genève-Bout-du-Monde (46.18028, 6.15937, FOEN) | Station at Genève-bout-du-monde (46.180332, 6.159276, SIG/INRAE) 46.180332, 6.159276 |
| Fier (1 380 km², 41 m³ s⁻¹, NA) | Steep mountain torrent in the Pre-Alps with high flow in spring. | Permanent | Station V1264021 at Motz (45.9333845, 5.8415226, CNR) | Station at Motz (45.9333709, 5.8411925,INRAE) 45.9333709, 5.8411925 |
| Guiers (617 km², 16 m³ s⁻¹, NA) | Steep mountain torrent in the Pre-Alps with high flow in spring. | Temporary | Station V1534020 at Belmont-Tramonet (45.5724539, 5.6520427, CNR) | Station at Belmont-Tramonet (45.5724539, 5.6520427, INRAE) |
| Bourbre (728 km², 8 m³ s⁻¹, NA) | Agricultural tributary with high flow in winter. | Temporary | Station V1774010 at Tignieu-Jameyzieu (45.7152493 , 5.1591093, DREAL Auvergne Rhône Alpes) | Station at Tignieu-Jameyzieu (45.7152493, 5.1591093, INRAE) |
| Ain (3 765 km², 130 m³ s⁻¹, NA) | Steep gravel-bed river with high flow in winter. One of the main tributaries in the upper Rhône River | Temporary | Station V2942010 at Chazey-sur-Ain (45.9063609, 5.2340163, DREAL Auvergne Rhône Alpes) | Station at Chazey-sur-Ain (45.9063609, 5.2340163, INRAE) |
| Saône (29 950 km², 416 m³ s⁻¹, 0.32 ± 0.12 Mt) | Large lowland river with high flow in winter. Major tributary of the upper Rhône River and one of the main providers of SPM flux. | Permanent | Station U4710011 at Couzon-au-Mont-d'or (45.8470505 , 4.8354978, CNR) | Station, at Lyon (45.757393, 4.825801, INRAE) 45.794294, 4.827073 |
| Gier (417 km², 3 m³ s⁻¹, 6.7 ± 7.3 10⁻³ Mt) | Minor tributary with high flow in winter. | Permanent | Station V3124010 at Givors (45.5795623, 4.7398048, DREAL Auvergne Rhône Alpes) | Station at Givors (45.5794239, 4.7389465, INRAE) |

| Isère (11 890 km², 309 m³ s⁻¹, 1.8 ± 1.5 Mt) | Mountainous tributary with high flow in spring. Major tributary of the upper Rhône River and one of the main providers of SPM flux | Permanent | Station W3540010 at Beaumont-Monteux (45.0165428, 4.9149795, CNR) | Station at Beaumont-Monteux (45.017036, 4.913693, EDF) | 45.00651, 4.89693 |
|---|---|---|---|---|---|
| Drôme (1 663 km², 20 m³ s⁻¹, NA) | Medium-sized mountainous tributary with high flow in spring. | Permanent | Station V428701201 at Livron-sur-Drôme (44.766044, 4.840190 - CNR) | 44.766044, 4.840190 - INRAE | |
| Ardèche (2 376 km², 65 m³ s⁻¹, 0.05 ± 0.03 Mt) | Medium-sized tributary with high flow in spring and fall . | Permanent | Station V5064010 at Saint-Martin-d'Ardèche (44.3139851, 4.5511069 - SPC Grand Delta) | 44.3139851, 4.5511069 - INRAE | 44.299977, 4.569605 |
| Durance (14 225 km², 68 m³ s⁻¹, 1.7 ± 0.4 Mt) | Mountainous tributary with high flow in spring. Major tributary of the lower Rhône River and one of the main providers of SPM flux | Permanent | Station X350001001 at Bonpas (43.8887991 4.9231973, CNR) | Station X350001001 à Bonpas (43.8887991 4.9231973, EDF) | 43.888843, 4.916630 |
| Gardon (2 040 km², 33 m³ s⁻¹, NA) | Minor tributary with high flow in spring and fall . | Permanent | Station V7194005 at Remoulins (43.9379640 4.5578986, CNR) | 43.940167, 4.5575068 - INRAE | 43.905728, 4.584211 |

### 3.2 Suspended Solid Concentration (mg L⁻¹)

The SSC at most stations are derived from in-situ turbidity measurements conducted every 10 minutes (Le Bescond et al., 2018). Universal Controller SC100 or SC200 (HachLange, Germany) are used in addition to numerical Solitax SC optical turbidity probes (Fig. 4A), all equipped with a mechanical cleaning system (wiper). Sensors use the infrared scattered light method with the optical response being dependent on sediment characteristics in the water. The turbidity meter is usually
immersed at a fixed position along the riverbank near the station, avoiding dead zones or effluents so that the measured turbidity is representative of the average turbidity throughout the river cross-section. Exceptions: at Jons, river water is pumped and circulated to an in-door turbidity meter; at Arles, there is no turbidity meter. The SSC is then calculated through the site-specific turbidity-SPM rating curve (Navratil et al., 2011), which is determined on each site for a wide range of
concentrations (Table 4). The curves are established using water samples collected manually or by automatic samplers (Fig. 4B) triggered hourly during flood events. Water samples are collected regularly to ensure there is no change in the relationship between turbidity and SPM. A new turbidity-SPM rating curve is systematically built when a turbidity probe is replaced. For the Isère, the Durance and the Andancette stations, the conversion is computed by the external provider (Table 2 and 3). In order to determine SPM values from these samples, they are filtered through pre-weighed glass fiber filters,

dried (105°C during 2 hours) and weighted according to the standard method NF EN 872 (AFNOR, 2005). Relative
       uncertainty on SPM concentrations is estimated to 9% (at 95% uncertainty level, coverage factor k=2).

Table 4 - Calibrations of turbidimeter and sampling

| Station | Scale set period | Turbidity/SSC coefficient | $R^2$ | Minimum SSC (mg L$^{-1}$) | Maximum SSC (mg L$^{-1}$) | Number of sample analyzed |
|---|---|---|---|---|---|---|
| Rhône River at Jons | 06/2011 – in progress | 1.09 | 0.959 | 1.6 | 950 | 752 |
| Arve | 06/2012 – 08/2013 | 1.12 | 0.944 | 5.0 | 9600 | 262 |
|  | 02/2015 – in progress | 0.80 | 0.902 | 3.0 | 12400 | 285 |
| Fier | 04/2014 – 06/2017 | 1.87 | 0.723 | 1.1 | 2860 | 179 |
|  | 08/2017 – 10/2020 | 1.08 | 0.965 | 0.2 | 1794 | 214 |
|  | 12/2020 – in progress | 0.97 | 0.985 | 5.0 | 1329 | 83 |
| Guiers | 04/2021 – 07/2012 | 0.38 | 0.92 | 0 | 668 | 21 |
| Bourbre | 10/2011 – 03/2012 | 1.09 | 0.95 | 31 | 314 | 20 |
|  | 02/2013 – 10/2013 | 0.56 | 0.60 | 9 | 84 | 17 |
| Ain | 07/2012 – 01/2013 | 0.56 | 0.91 | 1 | 124 | 45 |
|  | 05/2016 – 06/2017 | 1.17 | 0.79 | 1 | 85 | 21 |
| Saône | 01/2010 – 05/2012 | - | - | 9.3 | 71 | 18 |
|  | 02/2014 – 10/2020 | 0.92 | 0.91 | 1.3 | 192 | 200 |
|  | 12/2020 – 09/2021 | 1.11 | 0.79 | 3.0 | 86 | 44 |
|  | 09/2021 – in progress | 1.07 | 0.91 | 3.5 | 75 | 59 |
| Gier | 04/2013 – 11/2019 | 1.08 | 0.891 | 1.2 | 1342 | 151 |
|  | 12/2019 – in progress | 1.28 | 0.81 | 2.5 | 775 | 83 |
| Drôme | 11/2018 – 08/2020 | 0.79 | 0.909 | 2.0 | 8728 | 227 |
|  | 05/2021 – in progress | 1.05 | 0.98 | 1.0 | 833 | 52 |
| Ardèche | 01/2016 – in progress | 1.43 | 0.866 | 1.0 | 340 | 63 |
| Gardon | 06/2017 – 11/2020 | - | - | 0.6 | 3.8 | 6 |
|  | 05/2021 – in progress | 0.54 | 0.64 | 0.6 | 5.4 | 7 |

At Arles, near the outlet of the Rhône River to the Mediterranean Sea, SSC are measured by the MOOSE network
       (Mediterranean Ocean Observing System for the Environment) with sampling conducted in the SORA monitoring station
       (Raimbault et al., 2014). Water intake is located on a floatable structure at a distance of 7 m from the bank and 0.5 m under
       the surface. Sampling for SSC is achieved using a cooled automatic water sampler that fills a daily bottle with 150 mL every
       90 minutes (Eyrolle et al., 2010). During flood events (water discharge greater than 3000 m$^3$ s$^{-1}$), 150 mL samples are

180 collected every 30 minutes to constitute a composite sample every 4 hours. Samples are poisoned with $HgCl_2$ and kept at 5°C until they are filtered on GF/C Whatman pre-conditioned glass fiber filters (dried at 500°C for 4 hours). The filtered volumes are adapted to the charge of SPM.

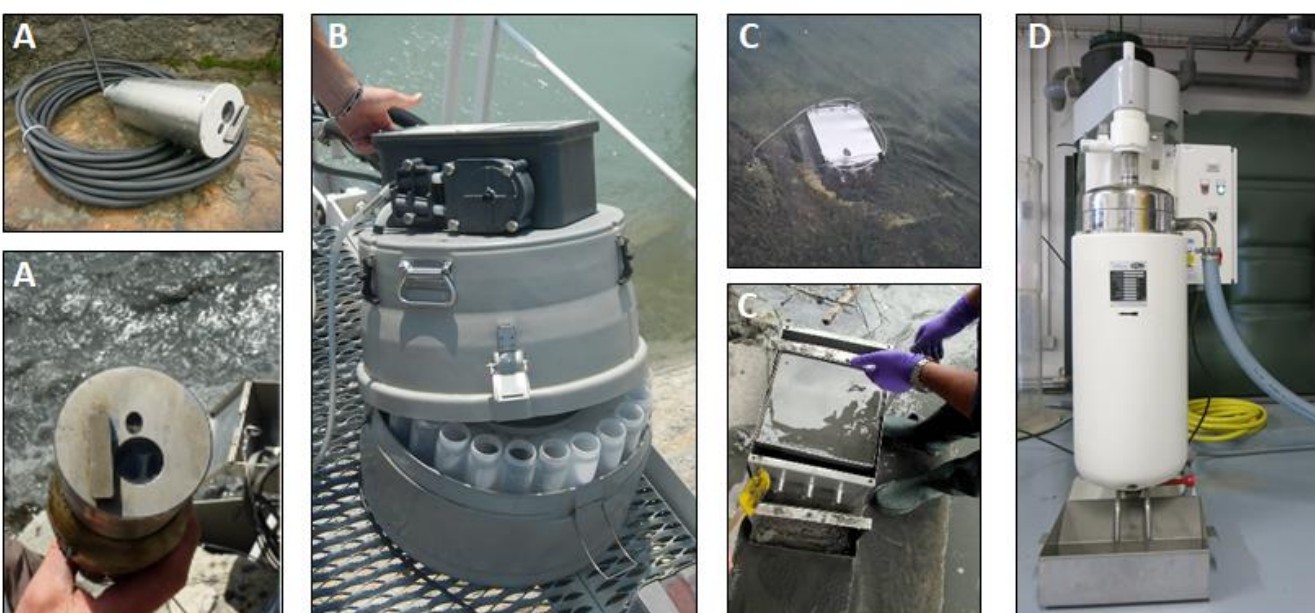

**Fig. 4 - Devices used for SSC measurement and SPM sampling. A) Hach Lange Solitax SC turbidity probes equipped with a wiper,**
**B) Automatic sampler (ISCO, Teledyne ISCO, Lincoln, USA) used for the calibration of turbidity stations, C) Particle Trap, and**
**D) Continuous Flow Centrifuge at Jons station.**

### 3.3 Sampling of SPM for analysis

The SPM used to measure POC, PSD and contaminant concentrations are mainly collected using particle traps (PT, Fig. 4C), while continuous flow centrifuges (CFC, Fig. 4D) are used to monitor specific events such as fast flood event. The PT are

190 rectangular stainless-steel boxes, whose internal flows circulate in two distinct parts separated by plates (Schulze et al., 2007; Masson et al., 2018). Such integrated sampling allows the collect of sufficient amounts of SPM for contaminants analysis. The PT are immersed near the riverbank (Fig. 4C) avoiding dead zones or effluents so that the sampled material is representative of the river fine suspension throughout the cross-section. For Andancette and the Saône river monitoring stations, the PT are suspended from a chain and kept immersed at a depth of 0.5 - 1 m while at the other stations the PT are

195 attached to the riverbed at an average depth of 0.5 m. At Arles, the PT and CFC are located inside the SORA monitoring station (Eyrolle et al., 2010) and supplied by a pipe. These devices allow to take into account the fluctuations of the SPM flux (Le Bescond et al., 2018; Delile et al., 2020). The PT is generally collected every month but can be collected at shorter time intervals in order to monitor specific events such as a flood or dam regulation. Unfortunately, PT were sometimes not

recovered due to logistic constraints including high level of water or vandalism. The measurements conducted on PT
samples are considered as time-averaged over its sampling period. The purpose of the PT is to obtain an integrative response
over a period, which does not allow for the assessment of variation that may occur within that sampling period.

In addition, SPM samples for contaminants analysis are occasionally collected using a high speed CFC (Masson et al., 2018),
especially for the TME at Arles and at Jons. The duration of pumping can be regulated from 10 min to 8 hours in order to
collect sufficient amount of SPM. No significant differences in Hg and PCB concentrations were found in samples collected
by the two methods although particles in PT are slightly coarser than particles collected using CFC (Masson et al., 2018).
The analyses are carried out on the total samples without separation of the organic part because it is negligible in the samples
(see the POC measurements). Prior to chemical analysis, SPM collected with the two sampling techniques are transferred to
Clean brown glass bottles (250 mL) are used to transfer the SPM. Prior to chemical analysis, samples were deep-frozen (-
18°C) and freeze-dried before being homogenized in an agate mortar and stored in the dark at ambient temperature.
In addition, excess SPM samples are stored in a chamber at -80°C. This will allow, according to the needs and the
development of new analytical techniques, to carry out later analysis without aging of the samples. More than 1300 samples
are currently stored this way. Meta-data and location of the samples inside the chamber are saved within the software Collec-
Science (Quinton et al., 2020).

### 3.4 Physico-chemical analyses

All the physico-chemical analyses information are briefly recalled during visualization and in an additional file (Report.txt)
when downloading the data.

### 3.4.1 Particle size distribution (PSD)

The PSD is measured by laser diffraction with a Cilas 1190 particle size analyzer (Cilas SA, Orléans, France). The
volumetric particle size distribution of SPM (measuring range: 0.04 to 2500 µm) was assessed by INRAE-RiverLy Aquatic
chemistry laboratory (AFNOR, 2009). During measurement (obscuration rate typically 15%), mechanical agitation in the
tank (at 350 rpm) and circulation with a peristaltic pump (at 120 rpm) were used in order to homogenize the sample. A
refractive index in the range of kaolinite was used for the solid phase (RI=1.55). Ultrasounds were used during dispersion
and during measurement in order to avoid particle aggregation (20 seconds at 38kHz). The PSD was also measured without
ultrasound for comparison. The volumetric particle size distribution of the sample was computed using the Fraunhofer
optical model. A quality control sample (made by INRAE) was systematically used to control the device.

Before 2018, the PSD of Ardèche, Durance, Gardon and Arles stations were measured without ultrasounds by a Beckman
Coulter LS 13 320 (Beckman Coulter, Fullerton, CA, USA) at the CEREGE laboratory. Prior to measurement with this
device (measuring range: 0.04 to 2000 µm), the organic component of the water sample was oxidized using a solution of
30% hydrogen peroxide ($H_2O_2$) at 200 °C. The remaining fraction was then resuspended in a 0.3% hexametaphosphate
solution and sub-sampled under stirring (800 rpm) to match the optimal obscuration windows of the laser and of the light

polarization system, between 8 and 16 % and between 50 and 70 % respectively. The volumetric particle size calculation model was performed in accordance with the Fraunhöfer and Mie theory. A refractive index in the range of kaolinite was used for the solid phase (RI=1.56). Each sample was analyzed 6 times (90 seconds each) with water circulation at 80% and the result finally recorded is an average of the 5 last runs, because some air bubbles sometime alter the first run just after the rinsing phase. Sample size reproducibility based on independent replicate measurements of the same sample did not exceed 2% residual (Psomiadis et al., 2014).

The database provides information on characteristic diameters D10, D50 and D90 of the particle size distribution. Non negligible differences between these parameters were found during an intercomparison of the two devices (Lepage et al., 2019). The comparison of the results must therefore be done with caution. Finally, the entire information on particle size distribution of each sample is stored by INRAE and CEREGE laboratories.

### 3.4.2 Particulate organic carbon (POC)

At Arles, POC was measured in filtered water samples within the MOOSE observatory (Raimbault et al., 2014). The SPM was collected on GF/C Whatman glass fiber filters (25 mm in diameter) precombusted at 500°c during 4 hours. Filters were dried at 60°C and stored dried until analysis. Before analysis the filters were placed in tin capsules and acidified with sulfuric acid (0.25 N) to remove inorganic carbon, and dried at 60°C. The POC measurements were performed using high combustion procedure (950° C) on a CN Integra mass spectrometer (serCon Ltd, Crewe, UK) according to Raimbault et al. (2008). Reference materials (glycine and casein) were systematically used to control analytical uncertainty.

For all the other stations, the determination of POC before December 2014 was performed using a CHN Flash 2000 carbon analyzer (ThermoFisher-Scientific, USA) by INRAE-RiverLy Aquatic chemistry laboratory. Prior to analysis, samples were decarbonated using hydrochloric acid HCl (AFNOR, 1995). Depending on the POC concentration, the analytical uncertainty ranged between ~3% and ~6% (k=2) while the limit of quantification (LQ) was estimated to be 0.1 g kg$^{-1}$. After December 2014, a different device was used (CNS Flash 2000, ThermoFisher Scientific, USA) following manufacturer recommendations and the above method. Analytical accuracy (93%) and uncertainty (8%; k = 2) were controlled using a reference material (AGLAE, 15 M9.1; 40 g kg$^{-1}$) and the LQ was estimated to be 0.5 g kg$^{-1}$.

### 3.4.3 Trace metal elements (TME)

The following TME were measured at the LA-ICP-MS platform for elemental chemistry of CEREGE: Cd, Co, Cr, Cu, Ni, Pb, Zn and As (Delile et al., 2020). All laboratory materials used were acid-cleaned and all reagents were ultrapure grade compounds. Samples of 40-mg dry SPM were dissolved in a mixture of 4 mL $HNO_3$ (67%), 20 drops of $H_2O_2$ (35%), 3 mL of HCl (34%), and 0.5 mL of HF (47-51%) before being digested in an UltraWAVE Single Reaction Chamber (Milestone, Sorisole, Italy) at 170°C (10 min) and 250°C (10 min; P = 100 bar). The complete breakdown of the SPM samples was

verified by the absence of residues. Measures were conducted by ICP-MS (Nexlon 300X, PerkinElmer, USA) after dilution of the samples with ultrapure Milli-Q water (Merck). Calibration curves and rhodium solution was used as internal standard while reference sediments PTSD-3 and MESS-4 (Canadian Certified Reference Materials Project) were analyzed repeatedly. The relative uncertainties were 6.7%, 14%, 3.6%, 1.2%, 5.0%, 2.3%, 3.2%, 3.0% respectively for As, Cd, Co, Cr, Cu, Ni, Pb, Zn while their LQ were 0.5 mg kg$^{-1}$, 0.1 mg kg$^{-1}$, 0.05 mg kg$^{-1}$, 0.1 mg kg$^{-1}$, 0.1 mg kg$^{-1}$, 1 mg kg$^{-1}$, 1 mg kg$^{-1}$, 5 mg kg$^{-1}$.

The determination of total Hg in SPM was performed by INRAE-RiverLy Aquatic chemistry laboratory (Delile et al., 2020) using an automated atomic absorption spectrophotometer DMA 80 (Milestone, Sorisole, Italy), following the EPA method 7473 (US EPA, 2007) and Aquaref MA02 method (Cossa et al., 2013). Blanks were systematically checked to verify the absence of contamination during analyses. Analytical uncertainty (16%; k=2) and accuracy (94%) were systematically controlled using reference materials (IAEA 457, coastal sediment; IAEA 458, marine sediment) and the LQ was 10 μg kg$^{-1}$.

### 3.4.4 Polychlorobiphenyls (PCB)

The concentrations of seven indicator PCBi were analyzed by INRAE-RiverLy Aquatic chemistry laboratory as described in Delile et al. (2020): congeners 28, 52, 101, 118, 138, 153, and 180. Among 1.0 g dry weight of SPM was extracted with a mixture of cyclohexane/acetone 90:10 v/v then concentrated by evaporation and purified on a 1 g Florisil SPE cartridge. To avoid sulfur interferences, small amount of copper powder (10 mg mL$^1$) was added prior to gas chromatography analysis with a $^{63}$Ni electron capture detector (GC-ECD). Two columns were used to analyze the samples (RTX®-5 and RTX®-PCB) and the accuracy was checked via the analysis of a certified reference material (BCR 536) and intercomparison exercises. The LQs were estimated between 0.5 and 1 mg kg$^1$ depending on the congeners while the analytical uncertainties was determine using a sediment sample from the Bourbre River as no certified reference material exists for such low levels of PCBi in equivalent matrix. Analytical uncertainties were estimated to be 60% (k = 2) for concentrations lower than 3-times the LQ, and to be 30% (k = 2) for concentrations higher than 3-times the LQ.

### 3.4.5 Radionuclides (cesium-137, organically bound tritium and radiocarbon)

For $^{137}$Cs activity, the SPM samples were ashed and put into tightly closed plastic boxes (17 mL or 60 mL) for gamma-ray spectrometry measurements (20–60 g) using low-background and high resolution High Purity Germanium detectors at the IRSN/LMRE laboratory (Eyrolle et al., 2020). These measurements being performed under 17025 accreditation, the laboratory participates each year in proficiency tests organized mainly by the ALMERA network (Analytical Laboratories for the Measurement of Environmental Radioactivity) of the IAEA. Each sample was measured for 3 days to achieve detection limits around 0.5 Bq kg$^{-1}$ d.w. for $^{137}$Cs, after waiting for 30 days for the radioactive equilibrium of the Ra-226's

progeny. Efficiency calibrations were constructed using gamma-ray standard sources in a 1.15 g cm$^3$ density solid resin–
water equivalent matrix. Activity results were corrected for true coincidence summing (TCS) and self-attenuation effects
(Lefèvre et al., 2003). Measured activities, expressed in Bq kg$^{-1}$ (d.w.), are decay-corrected to the date of sampling. The
activity uncertainty (k=2) was estimated as the combination of calibration uncertainties, counting statistics, and summing
and self-absorption correction uncertainties.

Organically bound tritium (OBT in Bq kg$^{-1}$ dry) analysis was performed at the IRSN/LMRE laboratory by the Helium-3
(3He) ingrowth method (Cossonnet et al., 2009). The samples were put under vacuum (10$^{-6}$ mbar) and stored up to 4 months
prior to analysis. The 3He/4He ratio was measured by mass spectrometry after correction of the radiogenic 4He levels
contained in the sample (gaseous inclusions) and normalization of the values to ambient atmospheric levels (3He/4He).

Radiocarbon ($^{14}$C) contents was analyzed by the IRSN/LMRE laboratory using an accelerator mass spectrometer (LMC14
laboratory, Saclay, France). Prior to analyze the organic part of the sample, carbonates were eliminated by washing the
sample (0.5MHCl, 0.1MNaOH) and drying it under vacuum as describe in Eyrolle et al. (2018). Decarbonated samples were
then sealed in quartz tubes under a vacuum with an excess of CuO and silver wire. Organic carbon was converted into CO2
by introducing the tubes into a furnace at 835 °C for 5 h and then released, dried, measured, and collected after broking the
tube under a vacuum. The graphite target is obtained with a direct catalytic reduction of the CO$_2$, using iron powder as a
catalyst (Merck® for analysis reduced, 10 μm particles). The reduction reaction occurs at 600 °C with excess H2 (H2/CO2 =
2.5) and is complete after 4–5 h. The iron-carbon powder is pressed into a flat pellet and stored under pure argon in a sealed
tube. All quartz and glass dishes are burned for at least 5 h at 450 °C to reduce contamination. To evacuate the vacuum lines,
a turbo-molecular pump reaching 10$^{-6}$ mbar is used. Measurements are performed using a 3 MV NEC Pelletron Accelerator
coupled with a spectrometer dedicated to radiocarbon dating, measuring 12C, 13C and 14C contents and counting the 14C
ions by isobaric discrimination. Analysis require 1 to 100 mg of dry sample (to obtain 1 mg of carbon). The specific 14C
activity is expressed as Becquerel of 14C per kilogram of total organic carbon (Bq kg$^{-1}$ of C). The detection limit is 0.8 Bq
kg$^{-1}$ of C and the uncertainty is 0.1% for modern samples (k= 2).

### 3.5 Data completion and flux calculation

The SPM fluxes are the product of water discharges and SSC, and contaminant fluxes are the product of the SPM fluxes and
the contaminant concentrations (Fig. 1). At most stations, water discharge is provided by a collocated or neighbouring
hydrometric station. Most often, water levels are measured using pressure sensors, pneumatic probes (bubblers) or radar
gauges, and the stage records are converted to discharge using a stage-discharge rating curve (Le Coz et al., 2014; Kiang et
al., 2018), or a stage-fall-discharge rating curve (Mansanarez et al., 2016) for stations affected by variable backwater
upstream of a dam. Hourly averaged water discharge data are generally calculated by conversion of water level
measurements through stage-discharge rating curves, otherwise through numerical modelling. At Jons, the closest
hydrometric station is relatively far upstream, and two tributaries bring significant amounts of water between the

hydrometric and the turbidity station. Therefore, a 1-D hydrodynamic model (Dugué et al., 2015; Launay et al., 2019) is used to compute the discharge time series at Jons from the three discharge times series measured upstream on the Rhône River (at

330 Lagnieu) and on the two tributaries (Ain and Bourbre Rivers).

Prior to calculate these fluxes, completion of missing values (gaps in the measurement or non-monitored periods) and time step transformation are required.

### 3.5.1 Completion of missing values of SSC

Missing SSC values are estimated using the empirical relations between water discharges (Q in $m^3$ $s^{-1}$), and SSC (Cs in mg

$L^{-1}$) also known as sediment rating curves (Horowitz, 2003; Sadaoui et al., 2016). The Q-SSC relations (Eq. (1)) were improved by considering a low/moderate water discharges segment (a1 and c1) and a high water discharges segment (a2, b2 and c2) (Sadaoui et al., 2016):

$$Cs = a1\ Q^{c1} \qquad \text{if } Q<k \tag{1}$$
$$Cs = a2\ (Q-b2)^{c2} \qquad \text{otherwise}$$

The BaRatin method and the BaRatinAGE software (Le Coz et al., 2014) was used to fit the regressions and detects the breakpoints (k in $m^3$ $s^{-1}$), above which the regression coefficients change significantly. Discharge-SPM rating curves are too uncertain to allow the detection of potential temporal changes. We therefore assume that they are constant over the monitoring period. Estimated parameters of Eq. (1) are listed by station in table 4 according to Poulier et al. (2019) and updated in 2021 with new data. Estimated values take the code "e" (Table 1).

**Table 4 - Parameters of the discharge-SSC relations (Eq. (1)) for all the OSR monitoring stations.**

| Site | a1 | c1 | k | a2 | c2 | b2 |
|---|---|---|---|---|---|---|
| Rhône River at Jons | 0.00364161 | 1.27394 | 606 | 3.83E-06 | 2.37002 | 40.70914059 |
| Rhône River at Andancette | 0.10147 | 0.613183 | 901 | 0.00591612 | 1.26049 | 640.0302716 |
| Rhône River at Arles | 0.000140155 | 1.69708 | 2744 | 0.000306149 | 1.91365 | 1999.370307 |
| Ain River | 0 | 0 | 0 | 0.0002332 | 1.85508 | -117.77481 |
| Ardèche River | 0.999981 | 0.604738 | 259 | 0.000213873 | 1.73555 | -643.4083506 |
| Arve River | 0.130169 | 1.57866 | 320 | 0.00116687 | 2.00387 | -669.319577 |
| Bourbre River | 14.9267 | 0.628335 | 15 | 0.489962 | 1.92888 | 0.797376312 |
| Drôme River | 1.17261 | 1.39694 | 70 | 0.993729 | 1.5347 | 16.75046523 |
| Durance River | 0.269305 | 1.19943 | 311 | 1.06461 | 1.26776 | 233.8140539 |
| Fier River | 3.99998 | 0.178118 | 40 | 0.000339675 | 2.55091 | -11.02265988 |
| Gardon River | 1.01815 | 0.656233 | 98 | 0.874559 | 0.977631 | 72.64053285 |
| Gier River | 7.66621 | 0.821594 | 13 | 1.2623 | 1.21854 | -11.77340341 |
| Guiers River | 0 | 0 | 0 | 0.00785724 | 2.16924 | -8.523827187 |
| Isère River | 0.00563359 | 1.53469 | 450 | 0.00146434 | 2.15816 | 306.1673008 |
| Saône River | 0.00E+00 | 0.00E+00 | 0.00E+00 | 0.00173893 | 1.37465 | -279.3516091 |

### 3.5.2 Completion of missing values of contaminant

For PT, missing values are considered when two successive samplings are not continuous (i.e the start date of a sampling is later than the end date of the previous sampling). Prior to estimate missing values for PT samplings and when the samplings are successive, discontinuous time series are transformed to continuous time series by linear interpolation (Fig. 5 and 6). Missing values were replaced by the median value of the contaminant concentration depending on the hydrological conditions (baseflow or flood) during the integrated sampling period as described in Delile et al. (2020) (Fig. 5). In brief, samples were considered as taken in flood when more than 50% of the SPM cumulative flux occurred while the water discharge was higher than the flood threshold (defined as half of the 2-year flood peak discharge). For gap periods greater than the usual time period of sampling (28 days), the gap periods were split in two to avoid gap periods greater than 1.5 times this usual sampling period.

For CFC, missing values are considered when the period between two samplings is longer than the usual time period of sampling. For gaps shorter than the usual sampling period, discontinuous time series are transformed to continuous time series by considering the concentration of the last sample until the half of the gap period, while the other half was filled with the next sample concentration (Fig. 5). Like with PT, median values for hydrological conditions were calculated to estimate the missing values (Fig. 5). The hydrological condition of a sample was considered as flood if the mean daily water discharge value was higher than the flood threshold. For gaps longer than the usual sampling period, the same rule as for PT was applied.

For both methods, values lower than the LQ were replaced by this LQ divided by 2. Estimated values take the code "e" (Table 1).

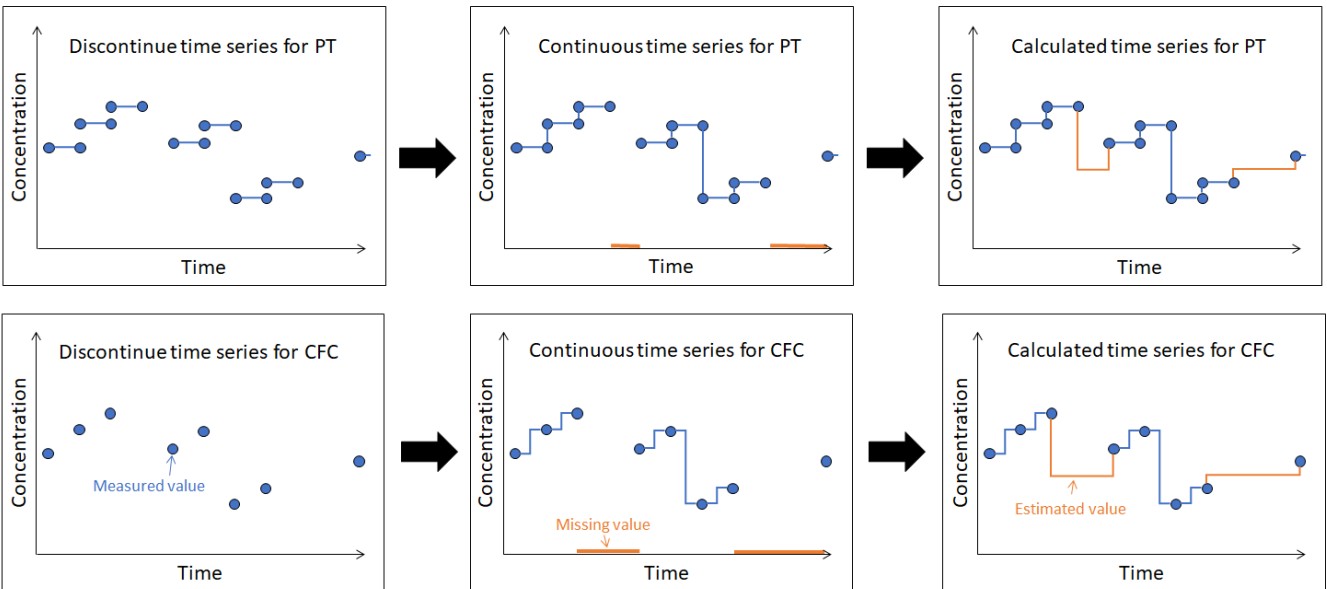

**Fig. 5 - Transformation step applied to a discontinuous time series for flux calculation. PT: particle trap, CFC: continuous flow centrifuge.**

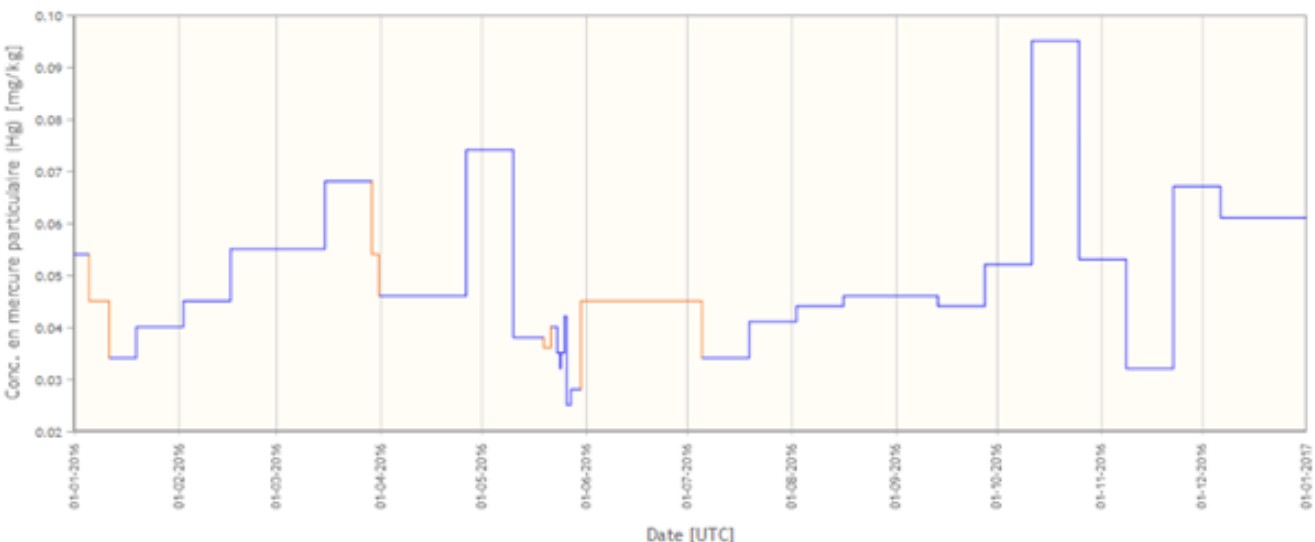

**Fig. 6 - BDOH/OSR database screenshot of the continuous time series with estimated concentration (orange) of Hg concentration at Jons in samples collected by particle trap.**

### 3.5.3 Time step transformation

After their completion, the time series are set to the same time step by linear interpolation to the nearest second between two points, so as not to lose any information. For samples collected by PT, the concentration measured is considered as a mean of

375 its sampling period. The SPM (g s⁻¹) and contaminant fluxes (g s⁻¹ or Bq s⁻¹) calculation is then carried out following Eq. (2) and Eq. (3), respectively. The lowest quality code of both values used is kept by respecting the following ranking: v>a>d>e.

$$FMES = Q * CMES * 1000 \text{ with Q in } m^3 \text{ s}^{-1} \text{ and CMES in g L}^{-1}. \tag{2}$$

$$FX = FMES * CX \text{ with CX the concentration of the contaminant X in } \mu g \text{ kg}^{-1}, \text{ mg kg}^{-1}, \text{ g kg}^{-1} \text{ or Bq kg}^{-1}. \tag{3}$$

### 4 Examples of applications using the dataset

With the data acquired by this network, it was possible to improve the calculation of the SPM and associated contaminant fluxes near the outlet of the Rhône River, and also to evaluate the fluxes coming from the upper Rhône River and the tributaries (Poulier et al., 2019; Delile et al., 2020). It was estimated that on average 6.6 Mt of SPM transited each year in the Rhône at Beaucaire, with strong variation ranging from 1.4 to 18.0 Mt y⁻¹. The Durance and Isère tributaries were found to be the main contributors to SPM fluxes.

Through this long-term continuous monitoring and its spatial resolution with sampling on the main tributaries, seasonal variations in several contaminant concentrations have recently been highlighted (Delile et al., 2020) as well as the impact of dam flushing operations (Lepage et al., 2020). Nearly two-thirds of the annual contaminant fluxes are released into the Mediterranean Sea during three short term periods over the year: 24% during a Mediterranean component in November, 15%

during oceanic rainfall component in January and 24% during nival component in May-June. During flushing operations
recorded from 2011 to 2016, the mean SPM concentrations were 6 to 8 times higher than during flood events at equal water discharge (Lepage et al., 2020).

At the Jons station, an original fingerprinting method was conducted based on the TME residual fraction in SPM (Dabrin et al., 2021). This approach demonstrated that under base flow conditions and during dam flushing operations, SPM originated mainly from the Arve River, while the origin was more contrasted during flood events.

Finally, within the OSR and using this database, a 1-D hydrodynamic model of SPM dynamics was developed (Launay et al., 2019). This model simulated the concentration of SPM at Arles during a flood event occurring in the Isère and the Durance tributaries. This model was also applied by Dabrin *et al.* (2021) for the fingerprinting approach and the combination of these two approaches demonstrated that a large portion of SPM from the Arve River was old sediment stored behind the Verbois dam and re-suspended during the dam flushing operation.

## 5 Data availability

All the data are made publicly available in French and English through the BDOH/OSR database (Thollet et al., 2021) at https://doi.org/10.15454/RJCQZ7 (Lepage et al., 2021). The BDOH (Base de Données pour les Observatoires en Hydrologie) application is managed by INRAE. The data is freely available for visualization, and for download after registration of a personal account.

## 6 Perspectives

The database presented in this paper will be continuously updated in the coming years, at least until the end of the current OSR program (2021-2024). Discharge, SSC, and particulate contaminants are continuously measured at the permanent stations and data are regularly updated online. The following improvements are planned:

– Uncertainties on the SPM and contaminants fluxes will be calculated and published;
– Other TME and new contaminants such as gadolinium will be included in the dataset depending on the authorities's concern and scientific purpose. These contaminants might also be analyzed on the SPM samples already collected and stored in the chamber at -80°C;
– Additional parameters to describe the particle size distribution will be added such as the percentage of clay/silt/sand;
– Link between the different contaminants measured on the same SPM sample will be added as a common sample name;
– The values lower than LQ or LD will be replaced by a more precise statistical method when feasible.

**Acknowledgments**

This study was conducted within the Rhône Sediment Observatory, a multi-partner research program funded through the Plan Rhône by the European Regional Development Fund (ERDF), Agence de l'eau RMC, CNR, EDF, and three regional councils (Auvergne-Rhône-Alpes, Région Sud - PACA, and Occitanie). The authors would like to thank the OSR staff for suspended particulate matter sampling and analytical measurement: H. Angot, J. Panay, C. Le Bescond, M. Lagouy, G. Grisot, L. Richard, S. Gairoard, J.C. Gattacceca, G. Dur, F. Eyrolle, F. Giner, D. Mourier, P. Paulat, J. Faramond, C. Antonelli, A. De vismes, C. Ardois, C. Le Corre, A. Bonnefoy, M. Fornier. We are very grateful to CNR, DREAL ARA, EDF, Grand Lyon, FOEN, SIG, SPC Grand Delta, Véolia and VNF for cooperating with OSR and sharing their monitoring data. A special thanks to the ZABR and the GRAIE for their interaction with the OSR and their help with logistics.

**Author contribution**

Hugo Lepage: Conceptualization, Data curation, Formal analysis, Funding acquisition, Methodology, Visualization, Writing – original draft preparation; Alexandra Gruat: Conceptualization, Data curation, Formal analysis, Methodology, Visualization, Writing – original draft preparation; Fabien Thollet: Conceptualization, Data curation, Formal analysis, Methodology, Visualization, Writing – original draft preparation; Jérôme Le Coz: Conceptualization, Formal analysis, Methodology, Funding acquisition, Writing – original draft preparation; Marina Coquery: Conceptualization, Formal analysis, Funding acquisition, Project administration, Writing – original draft preparation; Matthieu Masson: Formal analysis, Methodology, Writing – original draft preparation; Aymeric Dabrin: Formal analysis, Funding acquisition, Writing – original draft preparation; Olivier Radakovitch: Conceptualization, Formal analysis, Funding acquisition, Project administration, Writing – original draft preparation; Jérôme Labille: Funding acquisition, Writing – original draft preparation; Jean-Paul Ambrosi: Conceptualization, Funding acquisition, Writing – original draft preparation; Doriane Delanghe: Data curation, Methodology, Writing – original draft preparation; Patrick Raimbault: Funding acquisition, Writing – original draft preparation.

**Competing interests**

The authors declare that there is no conflict of interest.

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
