# Peer review of "Concentrations and fluxes of suspended particulate matters and associated contaminants in the Rhône River from Lake Geneva to the Mediterranean Sea"

_Earth System Science Data, 2021_

## Author Response (AR1)

**Response to the Referees and marked-up manuscript**

**1. Referee 1**

| Comment | Author's response | Author's change in manuscript |
|---|---|---|
| The authors provide information about a very interesting database. In general, the content of the article is good enough. It is advisable to make small changes. | | |
| 1. The quality of Figure 2 may be improved | Quality has been improved | new figure |
| 2. It is necessary to add to Figure 3 the location of all dams, as well as cooling ponds of nuclear power plants. | Added. | new figure |
| 3. It is not clear why to mention Geneva as a large city if the sediment runoff from Lake Geneva is taken as insignificant. | The city of Geneva is located downstream the Lake Geneva, and might be a source of particles or pollutants. Thus, we have not modified the manuscript. | |
| Section 3.2 Suspended Solid Concentration (mg L$^{-1}$) | | |
| 1. It would be useful to show the cross-sectional profiles of the river channels at the sampling sites and show the sampling points on these profiles. | The cross-sectional profiles are not yet acquired. Although the choice of the sampling site is indeed a crucial point for the fine suspension that we follow by turbidimeter, the problems of spatial homogeneity do not come so much from the shape of the section (which is simple on our sites) nor from its evolution, but from possible upstream contributions which would not have had time to mix: discharges from banks or tributaries. This was taken into account in the choice of stations and also in the position of the sensors or sampling points (not in unmixed areas on the bank or dead water areas). We added additional information on the manuscript. | The turbidity meter is usually immersed at a fixed position along the riverbank near the station, avoiding dead zones or effluents so that the measured turbidity is representative of the average turbidity throughout the river cross-section |

| | | |
|---|---|---|
| 2. The turbidity measurements by the optical method were carried out at one point of the channel? How were this measurement point chosen at each station? | The turbidity meter is usually immersed at a fixed position along the riverbank near the station, avoiding dead zones or effluents so that the measured turbidity is representative of the average turbidity throughout the river cross-section. Additional information was added | The turbidity meter is usually immersed at a fixed position along the riverbank near the station, avoiding dead zones or effluents so that the measured turbidity is representative of the average turbidity throughout the river cross-section. Exceptions: at Jons, river water is pumped and circulated to an in-door turbidity meter; at Arles, there is no turbidity meter. |
| 3. Text between lines 140 and 145. That is, the organic suspension was not separated from the mineral suspension. It is necessary to explain why this was done? | As the organic part of our samples is very low (mostly lower than 5%), the analyses were carried out on the total samples. Additional information was added | The analyses are carried out on the total samples without separation of the organic part because it is negligible in the samples (see the POC measurements). |
| 4. In the mouth of the river (Arles, near the outlet of the Rhône River to the Mediterranean Sea), sampling was carried out in 150 mL bottles. From what depth was the selection carried out? Did the sampling site change depend on the water flow rate? In what part of the riverbed was the sampling carried out? | The sampling is conducted inside the SORA Monitoring Station, that is supplied with water using a pump and a floatable structure at a distance of 7 m from the bank and 0.5 m under the surface regardless the water discharge (as previously explained L165). We have moved this explanation for clarity. | At Arles, near the outlet of the Rhône River to the Mediterranean Sea, SSC are measured by the MOOSE network (Mediterranean Ocean Observing System for the Environment) with sampling conducted in the SORA monitoring station (Raimbault et al., 2014). Water intake is located on a floatable structure at a |

| | | distance of 7 m from the bank and 0.5 m under the surface. Sampling for SSC is achieved using a cooled automatic water sampler that fills a daily bottle with 150 mL every 90 minutes (Eyrolle et al., 2010). |
| --- | --- | --- |

**2. Referee 2**

| Comment | Author's response | Author's change in manuscript |
| --- | --- | --- |
| The paper is interesting and present significant effort in comprehensive sediment-related studies – very outstanding Rhône Sediment Observatory (OSR) operated along Rhone river basin. I would expect that it can be published after revision. The main concern regarding various parts of the manuscript are presented below. | | |

| | | |
|---|---|---|
| Abstract. In the present form is to vague. It contains extensive general information (e.g. "suspended particulate matters (SPM) have been involved in the fate of hydrophobic contaminants such as polychlorobiphenyls (PCB), mercury (Hg) and other trace metal elements (TME), and radionuclides for decades") which are not in line with manuscript subject. Key results of the study should be additionally presented in the abstract. | Abstract has been rewritten | See abstract |
| Introduction. The overview of the sediment-related studies over Rhine basin are not fully substantive. Sediment budget studies and long-term changes in sediment budget along Rhone river and changes in sediment contamination due to environmental practices (is worth to discuss. Floods impact on sediment transport (https://doi.org/10.1016/j.geomorph.2007.06.00) was also well-known over Rhone river. It is quite important to illustrate them to emphasize gaps of knowledge and needs to maintain observatory. The temporal resolution of the observations should be mentioned here. | Additional information was added | See introduction |
| At the end of the Introduction section it is worth to present the main challenges of the OSR. How does OSR expands to the existed sediment monitoring network? What additional knowledge this monitoring network provides? | Additional information was added | In this watershed, studies conducted on sediment dynamics and associated contaminants are unfortunately scarse (Antonelli et al., 2008; Radakovitch et al., 2008; Panagiotopoulos et al., 2012; Delmas et al., 2012) and do not allow to understand the observed changes over the long term. On this basis, the monitoring of spatial and temporal |

| | | distribution of SPM and associated contaminants in the Rhône River has been conducted within the Rhône Sediment Observatory (OSR) since 2009 (Le Bescond et al., 2018). |
|---|---|---|
| 3.2. Suspended Solid Concentration. It is important to demonstrate site-specific turbidity-SPM rating curves (for each station – both significance of the relationships and explain possible spatial (and temporal differences). How often the relationships are recalibrated for each station? What is Relative uncertainty on SPM concentrations (9%) – how does estimate was received? | Sampling collection is carried out regularly - and never stopped - to ensure there is no change in the relationship between turbidity and SPM. A new turbidity-SPM rating curve is systematically begun when a turbidity probe is replaced. We added a table with the parameter of the rating curve and modified the manuscript. | The SSC is then calculated through the site-specific turbidity-SPM rating curve (Navratil et al., 2011), which is determined on each site for a wide range of concentrations (Table 4). The curves are established using water samples collected manually or by automatic samplers (Fig. 4B) triggered hourly during flood events. Water samples are collected regularly to ensure there is no change in the relationship between turbidity and SPM. A new turbidity-SPM rating curve is systematically built when a turbidity probe is replaced. For the Isère, the Durance and the Andancette stations, the conversion is computed by the external provider (Table 2 and 3). |

| | | |
|---|---|---|
| 3.3 Sampling of SPM for analysis. This section doÑƒÑ‹ not clearly demonstrate the frequency of the sampling. How many samples per year are taken? It is work to depict sampling periods charted on the hydrograph of representative station. | As explained in manuscript L161, on average, one sample per station is collected each month, but this number may be higher due to the occurrence of flood events, or lower due to logistical constraints or vandalism. Because of the long time period of sampling and the many stations, it is complicated to summarize the number of samples. However, this information is easily found on the BDOH database tool for a contaminant/station pair.
The BDOH database tool allows the user to view both the sampling period of a contaminant (in terms of its concentration) and the water discharge. | no change in the manuscript |
| Explain how PT is installed into the flow. | For most of the stations, the PT are immersed at an average depth of 0.5 m near the riverbank avoiding dead zones or effluents so that the sampling is representative of the river cross-section. For Andancette and the Saône river monitoring stations, the PT are kept submerged with chains at a depth of 0.5 - 1 m. . At Arles, the PT and CFC are located inside the SORA monitoring station and supplied by a pipe. Additional information was added | The PT are immersed near the riverbank (Fig. 4C) avoiding dead zones or effluents so that the sampled material is representative of the river fine suspension throughout the cross-section. For Andancette and the Saône river monitoring stations, the PT are suspended from a chain and kept immersed at a depth of 0.5 - 1 m while at the other stations the PT are attached to the riverbed at an average depth of 0.5 m. At Arles, the PT and CFC are located inside the SORA |

| | | monitoring station (Eyrolle et al., 2010) and supplied by a pipe. |
|---|---|---|
| It is not clear how the sampling procedure and frequency reflect high temporal variability of sediment contamination during floods (see 3.5.2 Completion of missing values of contaminant). This topic should be significantly elaborated. | We are aware that the PT is not designed here to evaluate the variation within an extreme event. The purpose of the PT is to obtain an integrative response of the event. In case we need to study a specific event, the centrifugation might be used to investigate the temporal variation within. We added additional information in the manuscript to clarify this part. Regarding the completion of the missing values, it is also true that the variation is not taken account. | The measurements conducted on PT samples are considered as time-averaged over its sampling period. The purpose of the PT is to obtain an integrative response over a period, which does not allow for the assessment of variation that may occur within that sampling period |

| | | |
|---|---|---|
| 3.5 Data completion and flux calculation. It is know clear the procedure of water discharge calculation. What data is used to count stage-discharge rating curves, how does water level observations are operated. What is numerical modelling used to count water discharges? | Discharge data are produced by others (private and public companies - mentioned in the BDOH database) and provided to us for data storage and flux computation. Additional information was added | At most stations, water discharge is provided by a collocated or neighboring hydrometric station. Most often, water levels are measured using pressure sensors, pneumatic probes (bubblers) or radar gauges, and the stage records are converted to discharge using a stage-discharge rating curve (Le Coz et al., 2014; Kiang et al., 2018), or a stage-fall-discharge rating curve (Mansanarez et al., 2016) for stations affected by variable backwater upstream of a dam. Hourly averaged water discharge data are generally calculated by conversion of water level measurements through stage-discharge rating curves, otherwise through numerical modelling. At Jons, the closest hydrometric station is relatively far upstream, and two tributaries bring significant amounts of water between the hydrometric and the turbidity station. Therefore, a 1-D hydrodynamic model (Dugué et al., 2015; Launay et al., 2019) is used to compute the discharge time series at Jons from the three discharge times series measured upstream on the Rhône River (at Lagnieu) and on the two tributaries (Ain and Bourbre Rivers). |

| | | |
|---|---|---|
| The possible temporal changes in sediment rating curves should be explained. How often relationships presented in table 4 are recalibrated? | Additional information was added | Discharge-SPM rating curves are too uncertain to allow the detection of potential temporal changes. We therefore assume that they are constant over the monitoring period. |
| The study should at the end compare the OSR network and system with similar initiatives Worldwide which provide comprehensive hydrogeochemical studies of large rivers sediment transport (e.g. ArcticGRO – see e.g. https://doi.org/10.1017/cbo9781139136853.026 ; ArcticFLUX – see https://doi.org/10.5194/acp-19-1941-2019 and  https://doi.org/10.24057/2071-9388-2018-11-1-6-19 ; | What the scientific literature tells us is that the more observatories we have of environmental variables (aquatic or otherwise), the better we are at understanding the global and complex processes that affect our environment. The work cited here concerns the Arctic environment and does mention the importance of monitoring freshwater inflow from rivers. We added information in the manuscript to mention this part and the need to develop monitoring programs and the sharing of their data. | See introduction |